# C/EBPβ Coupled with E2F2 Promoted the Proliferation of hESC-Derived Hepatocytes through Direct Binding to the Promoter Regions of Cell-Cycle-Related Genes

**DOI:** 10.3390/cells12030497

**Published:** 2023-02-02

**Authors:** Shoupei Liu, Jue Wang, Sen Chen, Zonglin Han, Haibin Wu, Honglin Chen, Yuyou Duan

**Affiliations:** 1Laboratory of Stem Cells and Translational Medicine, Institutes for Life Sciences, School of Medicine, South China University of Technology, Guangzhou 510006, China; 2School of Biology and Biological Engineering, South China University of Technology, Guangzhou 510006, China

**Keywords:** human embryonic stem cells, liver enriched transcription factor, C/EBPβ, E2F, hepatocyte differentiation, cell cycle

## Abstract

Human embryonic stem cells (hESCs) hold the potential to solve the problem of the shortage of functional hepatocytes in clinical applications and drug development. However, a large number of usable hepatocytes derived from hESCs cannot be effectively obtained due to the limited proliferation capacity. In this study, we found that enhancement of liver transcription factor C/EBPβ during hepatic differentiation could not only significantly promote the expression of hepatic genes, such as albumin, alpha fetoprotein, and alpha-1 antitrypsin, but also dramatically reinforce proliferation-related phenotypes, including increasing the expression of proliferative genes, such as CDC25C, CDC45L, and PCNA, and the activation of cell cycle and DNA replication pathways. In addition, the analysis of CUT&Tag sequencing further revealed that C/EBPβ is directly bound to the promoter region of proliferating genes to promote cell proliferation; this interaction between C/EBPβ and DNA sequences of the promoters was verified by luciferase assay. On the contrary, the knockdown of C/EBPβ could significantly inhibit the expression of the aforementioned proliferative genes. RNA transcriptome analysis and GSEA enrichment indicated that the E2F family was enriched, and the expression of E2F2 was changed with the overexpression or knockdown of C/EBPβ. Moreover, the results of CUT&Tag sequencing showed that C/EBPβ also directly bound the promoter of E2F2, regulating E2F2 expression. Interestingly, Co-IP analysis exhibited a direct binding between C/EBPβ and E2F2 proteins, and this interaction between these two proteins was also verified in the LO2 cell line, a hepatic progenitor cell line. Thus, our results demonstrated that C/EBPβ first initiated E2F2 expression and then coupled with E2F2 to regulate the expression of proliferative genes in hepatocytes during the differentiation of hESCs. Therefore, our findings open a new avenue to provide an in vitro efficient approach to generate proliferative hepatocytes to potentially meet the demands for use in cell-based therapeutics as well as for pharmaceutical and toxicological studies.

## 1. Introduction

Liver disorders are an increasingly intractable problem worldwide, drug-related hepatotoxicity accounts for more than 50% of acute liver failure cases, and liver transplantation is still the only treatment for patients with advanced liver failure [1]. In addition, drug-induced liver injury is also a major cause of drug discontinuation before and after the launch of newly developed drugs [2]; therefore, the development of the therapeutics and the research on mechanisms of liver disease heavily rely on the use of in vitro experimental models. Primary human hepatocytes (PHH) and their cultures are still considered the gold standard for liver-based in vitro models because they provide a good representation of in vivo conditions. In recent years, the development of in vitro models of PHH seems to be more suitable for liver disease modeling, such as hepatocytes being grown under certain conditions to form so-called 3D spherical structures [3]. Although the 3D spherical structure can inhibit dedifferentiation and maintain function for a relatively long period of time, the lack of donors limits the number of cells and their applications; moreover, isolated hepatocytes cannot effectively proliferate and expand in vitro [4,5], which also greatly limits the clinical and basic researches using hepatocytes.

In recent years, human embryonic stem cells (hESCs) may become an inexhaustible source of functional hepatocytes. Our previous studies have established a stable and effective strategy for the differentiation of hESCs into hepatocytes [6,7], and more recently, we established a hepatocyte polarization system, which significantly improved the function and drug metabolism activities of hESC-derived hepatocytes [8]. However, consistent with others’ reports, hESC-derived hepatic progenitor cells/hepatocytes with a limited capacity for proliferation have hindered the development of hepatocellular therapy, bioartificial liver devices, and drug screening and development. In order to obtain reliable, economical, and adequate hepatocytes, recently Li et al. [9,10] reported that, by adding a cocktail of chemical reagents coordinated the regulation of signaling pathways to optimize the proliferation, the number of iPSC-derived hepatic cells significantly increased, even single hepatoblasts were able to proliferate and form colonies, and the number of hepatoblasts could be increased nearly 3000 times within 3 weeks; meanwhile, the bipotency was maintained, which provided a reliable method for long-term stability and large-scale expansion of hepatoblasts. Although the approach of cell proliferation at the stage of hepatic progenitor cells has been developed, the specific molecular regulatory mechanisms involved remain unclear.

The transcription factor C/EBPβ is a member of the CCAAT enhancer binding protein family; C/EBPβ has been reported to play a crucial role in differentiation, energy metabolism, cell cycle, migration, and invasion of a variety of cells [11,12,13,14]. In the liver, C/EBPβ has been reported to be involved in murine liver circadian autophagy and liver regeneration in hepatectomy mice [15,16]. Initially, C/EBPβ was identified as a potential autophagy activator, which was rhythmically expressed in the liver and regulated by circadian and nutritional signals, further indicating that C/EBPβ regulated the temporal orchestration of circadian autophagy rhythm [15]. Secondly, in partial hepatectomy mice, the expression of C/EBPβ was increased rapidly and participated in the whole process of liver regeneration, which further revealed the specific regulation of cell-cycle genes involved in the process of liver regeneration [16]. In pluripotent stem cell (PSC)-derived hepatocytes, C/EBPβ has also been reported to be an important transcription factor in hepatic differentiation, and C/EBPβ mediates the expression of TGFBR2 to regulate and determine the fate of hepatoblasts [17]. In addition, C/EBPβ plays an important role in the proliferation and differentiation of various tissue and cell types. In the hematopoietic system, C/EBPβ is required for stress-induced granulocytogenesis in hematopoietic stem/progenitor cells. C/EBPβ^−/−^ hematopoietic stem/progenitor cells showed impaired cell-cycle activation at the early stage and delayed myeloid differentiation at the late stage in regeneration [13]. C/EBPβ also plays an important role in regulating lymphocyte growth and differentiation. After silencing C/EBPβ in multiple myeloma cell lines and primary multiple myeloma cells, cell proliferation was strongly inhibited, which was involved in the regulatory network of transcription factors essential for plasma cell differentiation, growth, and proliferation [18]. In adipocytes, C/EBPβ is required for mitotic clonal expansion (MCE) of 3T3-L1 precursor adipocytes, and histone demethylase Kdm4b and cell-cycle genes CDC45L, MCM3, GINS1, and CDC25C were identified as potential C/EBPβ target genes in MCE process [19]. In osteosarcoma, C/EBPβ acts as a transcription factor to inhibit the proliferation of osteosarcoma cells by regulating the expression of CLEC5A [20]. These results indicate that C/EBPβ is essential for the cell cycle and proliferation of a variety of cell types in mammals. Although C/EBPβ was reported to be involved in the differentiation of human PSCs, the specific roles and potential regulatory mechanisms of C/EBPβ in the process of PSCs-derived hepatocytes are still unclear.

In this study, in order to elucidate the role and regulatory mechanism of C/EBPβ in hepatic differentiation, the enhancement or knockdown of C/EBPβ was performed during the period of hepatic differentiation of hESCs. Initially, we aimed to improve hepatic differentiation by enhancing the expression of C/EBPβ; as expected, overexpression of C/EBPβ could significantly elevate hepatic phenotype. Intriguingly, we were surprised to find that the expression of proliferative genes such as CDC45L, CDC25C, and PCNA was significantly upregulated after enhancing the expression of C/EBPβ in the stage of hepatic progenitors. Afterward, RNA transcriptome analysis was performed to analyze differentially expressed genes (DEGs) and enrichment of signaling pathways after the overexpression/knockdown of C/EBPβ, and CUT&Tag sequencing was used to determine the interaction between C/EBPβ and DNA sequences, and Co-IP protein pull-down assay was further performed to investigate the protein interaction between C/EBPβ and its target. Finally, LO2 cells, a hepatic progenitor cell line, were used to verify the findings of the mechanism.

## 2. Materials and Methods

### 2.1. Hepatocyte Differentiation from hESCs

Human embryonic stem cell, H9 line, was purchased from WiCell Research Institute (Madison, WI, USA) under a Materials Transfer Agreement (No. 19-W0512), cultured and maintained on the embryonic fibroblast feeder layer (MEF) of CF-1 strain mice according to provider’s instructions. Differentiation of embryonic stem cells into hepatic progenitors underwent two critical stages under our culture condition, definitive endoderm (DE) was initially induced for 2 days with RPMI1640 medium supplemented with ActivinA (100 ng/mL) and Wnt3a (25 ng/mL), and then, fresh RPMI 1640 medium was replenished with 100 ng/mL Activin A, 0.5 mM sodium butyrate and 1 × B27 supplement for another 5 days. Afterward, the differentiation of hepatic progenitor cells was performed for 14 days under complete hepatocyte differentiation medium (HDM) containing IMDM medium supplemented with 20% FBS, 0.3 mM 1-thioglycerin, 0.5% DMSO, 100 nM dexamethasone, 0.126 U/mL insulin, HGF (20 ng/mL), FGF4 (20 ng/mL), BMP2 (10 ng/mL) and BMP4 (10 ng/mL) (all growth factors from Peprotech).

### 2.2. Flow Cytometry

The cells at the stage of differentiation were detected by flow cytometry to determine the gene expressions of different cell lineages. Briefly, 0.05% trypsin was used to dissociate the cells into single cells, and specific antibodies were labeled for flow cytometry detection and analysis. For the analysis of the surface markers, the cells were directly labeled with antibodies, and for the detection of the intracellular proteins, the cells were labeled with antibodies after the cells were fixed and permeabilized by a commercial Fixation/Permeabilization kit. Specific markers SSEA-4, TRA-1-81, CXCR4, SOX17, AFP, ALB, and A1AT were used to determine the phenotypes of hESCs and DE cells as well hepatic progenitor cells (HPCs). Antibody information is listed in Appendix A.

### 2.3. Real-Time Quantitative PCR Assay

Real-time quantitative PCR was used to detect the relative gene expression of cells at all stages of differentiation and after the knockdown or overexpression of C/EBPβ in hepatic progenitor cells. Briefly, total RNA was extracted by Trizol and then reversely transcripted into cDNA for subsequent qPCR experiments. The primers used are listed in Appendix A.

### 2.4. Protein Extraction and Western Blot

Total protein was extracted from differentiated hepatic progenitor cells with or without C/EBPβ knockdown/enhancement using RIPA lysis solution (containing 1 × protease inhibitor, 1 × phosphatase inhibitor, and 1 mM PMSF). BCA kit was then used to define protein concentration. Subsequently, SDS-PAGE agarose gel electrophoresis and Western blot analysis were performed according to manufacture. Primary and secondary antibodies were used in Appendix A.

### 2.5. Isolation and Culture of Human Primary Hepatocytes

The present study was approved by the Research Ethics Committee of Guangzhou First people’s Hospital (Ethical approval No.: K-2019-167). Primary liver cancer tissues and adjacent normal tissues were provided by the volunteers at Guangzhou First People’s Hospital, and a modified two-step collagenase perfusion procedure was used to isolate hepatocytes from normal tissue adjacent to cancer [21]. The isolated hepatocytes were seeded on cell culture plates coated with collagen type I-coated (Corning, Rat tail collagen type I, REF: 409-01, Corning, NY, USA) at a density of 1.5 × 10^5^/cm^2^ in commercial medium HCM (Lonza, CC-3198, Rochester, NY, USA) for further analysis.

### 2.6. Analysis of ALB Secretion by ELISA

The supernatant of differentiated hepatocytes cultured at different time points was collected, and the content of human ALB in the supernatant was determined by ELISA. Briefly, the well plate was first incubated with coated antibody, and then the diluted supernatant samples were added after 1% BSA blocking, followed by the addition of capturing antibody labeled with HRP. After that, the addition of TMB solution produced a color reaction, then a diluted concentrated sulfuric acid solution was used to terminate the reaction. The plate was read using a microplate reader (Biotek-800TS) with 450 nm wavelength for the detection and analysis of absorption. To calculate the ALB concentration, a powerful curve generation and analysis software, OriginPro 2019, was used for analysis.

### 2.7. Preparation of Lentivirus and siRNA Transfection

The Tet-On 3G systems are inducible gene expression systems for mammalian cells [22]. In Lenti-XTM Tet-On 3G inducible expression system, the Tet-On 3G protein undergoes a conformational change when doxycycline (DOX) is added, which allows it to bind to tet operator sequences located in the TRE3G promoter to initiate the expression of target genes. In our study, HPCs were induced to conditionally express C/EBPβ or sh-C/EBPβ under the control of a TRE3G promoter in the presence of DOX.

293T cell line was used to produce lentiviruses of plv-C/EBPβ and sh-C/EBPβ. The target plasmids were transduced into cells with PEI transfection reagent together, and package vectors of PsPA × 2 and PMD 2.G and the virus particles were produced and released into the culture supernatant. Then, supernatants containing virus particles were collected and mixed with 30% sucrose solution in a certain proportion, and the virus particles were obtained by ultra-centrifugation at 100,000× *g* for four hours. The precipitation was immersed with 100 uL of 1 × PBS at 4 °C for 24 h. The next day, the virus stock was acquired after resuspension, and the titer of the viruses was determined by ELISA (Lenti-X p24 Rapid Titer Kit, Takara, Kusatsu, Japan). After the detection of virus titers, HPCs were then transduced with lentiviruses plus polybrene (8 ug/mL). To maintain continuous inducible C/EBPβ or sh-C/EBPβ expression in culture, the medium was replaced and replenished with DOX (1 μg/mL, final concentration) every 24 h.

si-RNA (C/EBPβ) was synthesized by HanBio Technology Co., Ltd (Shanghai, China). The Oligo sequences of si-RNA (C/EBPβ) are listed in Appendix A. During the transduction, RNA-Fit reagent (HanBio, Shanghai, China) or Intefferin (Polyplus, REF: 409-01, Strasbourg, France) was used as a co-transduction reagent, and the formed transduction complexes were incubated in Opti-MEM™ Reduced Serum Medium (Gibco, Hangzhou, China). Then, the cell culture medium was changed to a basic medium IMDM (Gibco, Hangzhou, China) without serum and without penicillin–streptomycin–amphotericin B solution, and transduction complexes were added to the aforementioned medium. Six hours after transduction, the medium was changed to a complete hepatocyte differentiation medium HDM. Then, total RNAs were collected at 48 h after the transduction, and protein samples were collected at 72 h.

### 2.8. Analysis of RNA Transcriptome

Total RNA was extracted using Trizol, and RNA-Seq was performed at Beijing Novogene Technology Co., Ltd. (Beijing, China). RNA sequencing libraries were constructed using the Illumina mRNA-Seq Prep Kit. Fragmented and randomly primed 150 bp paired-end libraries were sequenced using Illumina HiSeq 4000. The above process of database construction was conducted by the Novogene. The raw data and clean data of D8, D12, D14, D16, D20, D24 was downloaded 24 December 2020, plv_Vector and plv_C/EBPβ was downloaded 28 February, 2022, si_NC and si_C/EBPβ was downloaded 8 January 2022, from Novogene Data Release platform: data-deliver.novogene.com.

### 2.9. CUT&Tag for Bench-Top Application

At the end of the differentiation of HPCs, 0.05% trypsin (Procell) was added to dissociate the cells, and single cells were harvested. The cell viability was determined by trypan blue staining, followed by treating with the standard operating procedure (SOP) of Hyperactive^R^ Universal CUT&Tag Assay kit for Illumina (Vazyme, TD903, Vazyme Biotech, Nanjing, China) to obtain the DNA fragments bound by the transcription factor C/EBPβ. Subsequently, the DNA library preparation and library amplification were conducted by TruePrep^R^ Index Kit V2 for Illumina (Vazyme, TD202). Then, the quality control (QC) and quantitative concentration detection were executed by Equalbit^R^ 1 × dsDNA HS Assay kit (Vazyme, EQ121) and Qubit™ 4.0 Fluorometer (Invitrogen™, Waltham, MA, USA). Finally, the resulting viable DNA libraries were sent out for sequencing and analysis at Guangzhou Gene denovo Technology Co., Ltd. (Guangzhou, China). The raw data was downloaded 12 July 2022, with subsequent data analysis using IGV software (IGV_2.14.1). A detailed, step-by-step protocol can be found at: https://www.protocols.io/view/bench-top-cut-amp-tag-wnufdew/abstract.

### 2.10. Co-Immunoprecipitation (CO-IP) Assays

Immunoprecipitation was conducted to reveal the protein interaction between C/EBPβ and E2F2 [23]. About 2 × 10^6^ cells were collected, and protein A/G agarose beads were used, which captured IgG from virtually all isotypes in cell culture supernatants and other antibodies for efficient antibody purification to remove nonspecific proteins in the first step. Then, the supernatants were incubated overnight at 4 °C with mouse anti-human C/EBPβ antibody (1:1000, santa cruz); afterward, protein A/G agarose beads were used again to catch C/EBPβ binding proteins and their protein conjugates. Subsequently, the aforementioned protein complexes were eluted from the beads, and then, agarose gel electrophoresis and Western blot were performed to transfer target proteins to PVDF membranes, followed by incubating with mouse anti-human C/EBPβ antibody (1:1000, santa cruz) or mouse anti-human E2F2 antibody (1:500, santa cruz). Next, the specific secondary antibody (Goat Anti-Mouse IgG HRP, 1:10,000, abmart), eliminating heavy chain interference, was used. Finally, the immunoblots were detected using an ECL detection system and quantified by Image J software (Version No.: 3.2.0.8).

### 2.11. Luciferase Reporter Assay

The human CDC45L promoter region (2000 bp upstream of TSS) containing C/EBPβ binding sites was synthesized, amplified, and cloned into the PGL3-basic luciferase vector at Guangzhou IGE Biotechnology Co., Ltd. (Guangzhou, China). Luciferase assays were performed by transiently transducing HEK293T cells or LO2 cells using Hexadimethrine bromide (Sigma, H9268, Saint Louis, MO, USA). Briefly, 5 × 10^4^ HEK293T cells or LO2 cells were seeded into 24-well plates; 24 h after the culture, cells were transduced with 1 μg of pGL3-CDC45L luciferase reporter constructs or pGL3-basic luciferase counterparts in each well together with 1 μg PLV-C/EBPβ plus DOX (1 μg/mL, final concentration). Cells were lysed 24 h post-transduction and analyzed for luciferase activity with the Luc-Pair™ Duo-Luciferase HS Assay Kit (GeneCopoeia, LF004, Rockville, MD, USA) according to the manufacturer’s protocol.

### 2.12. Statistics

All statistics were summarized for at least three independent measurements. An unpaired student *t*-test was used to analyze the data. *p* < 0.05 was considered statistically significant. *p* < 0.05 *, *p* < 0.01 **, *p* < 0.001 ***, and *p* < 0.0001 ****.

## 3. Results

### 3.1. C/EBPβ Was Enriched in Liver and Correlated with Hepatocyte Differentiation

Human pluripotent stem cells (hPSCs), including human embryonic stem cells (hESCs) or human iPS cells (hiPSCs), can be induced to form hepatocyte-like cells (HLCs) in a step-by-step process in vitro [24,25]. In our previous study, hepatocytes could be differentiated from hESCs by induction via sequential stages [6]. Briefly, hESCs mainly underwent three stages to differentiate into hepatocyte-like cells (HLCs), and the resulting HLCs showed typical polygonal characteristics in morphology (Figure 1A). To determine the process of differentiation, crucial markers were used to examine each stage. Flow cytometry results showed that hESCs specifically expressed stem cell markers OCT4 and SSEA4, DE cells co-expressed markers SOX17 and CXCR4 (Appendix A), and HPCs expressed hepatocyte markers AFP, ALB, and A1AT (Figure 1B). In HLCs, the expression of AFP, ALB, and A1AT was further enhanced, especially the ALB, a marker indicating cell maturity; both expression and secretion were significantly increased at the maturation stage (Figure 1C–E). However, there is still a significant difference in the expression level of ALB and ASGPR1 compared with primary hepatocytes (Figure 1C and Appendix A). The expression of these specific markers during each stage of differentiation was also verified by quantitative PCR (Appendix A).

Although the differentiated cells already exhibited hepatocyte phenotype, there is still a large gap in gene expression and hepatic function when compared to PHH. Previous studies have acknowledged the point that the enhancement of liver-enriched transcription factors (LETFs) could significantly improve hepatocyte gene expression and hepatic differentiation [26,27]. In our hepatic differentiation derived from hESCs, the expression of most LETFs was increased during the process of HPCs from day 8 to day 14, whereas a few transcription factors, such as ONECUT2 and C/EBPβ, maintained continuous increase throughout the process of differentiation (Figure 1G,H). Considering that ONECUT2 (also known as HNF6β) has been poorly studied in hepatocytes and is required for normal development of the biliary tract [28], the subsequent study focused on C/EBPβ. Several previous studies reported that C/EBPβ was involved in regulating the development and function of hepatocytes, including lipid homeostasis of hepatic autophagy [27] and liver regeneration and hepatocyte proliferation after partial liver resection [15]. In addition, the overexpression of C/EBPβ promoted hepatoblast differentiation by enhancing the expression of TGFβ-R2 [16]. In the online database, GO enrichment analysis in the KOBAS database shows a significant enrichment of signaling pathways related to C/EBPβ regulation, including hepatocyte proliferation, cell fate determination, and liver regeneration (Appendix A). In our results, C/EBPβ was closely related to hepatocyte differentiation; the expression trend of C/EBPβ had a strong positive correlation with several hepatocyte genes, such as AFP, ALB, ASGPR1, and CYP3A4 (Appendix A). The expression of C/EBPβ in HLCs was significantly increased when compared with those in HPCs; however, it was still far from that in PHH (Figure 1F). These results indicated that C/EBPβ was highly enriched in the hepatocytes and was closely related to hepatocyte differentiation.

### 3.2. Enhanced Expression of C/EBPβ Promoted Hepatic Differentiation

As a liver-enriched transcription factor, C/EBPβ was retained at a relatively low level in our differentiated hepatocytes, especially at the HPC stage. To further improve the hepatocyte differentiation, lentivirus-mediated transduction of C/EBPβ was conducted to express C/EBPβ in HPCs at the initial stage of the differentiation, and then, DOX was continuously added daily to induce the expression of C/EBPβ (Figure 2A,B). After the overexpression of C/EBPβ, as expected, we found that the total C/EBPβ expression was increased by about three times, and the exogenous C/EBPβ expression was increased by about six times in the HPCs that were transduced with lentivirus (Figure 2D). Consistently, the expressions of liver-specific genes such as AFP, ALB, ASGPR1, A1AT, HNF4α, and C/EBPα were all significantly increased (Figure 2E). The distribution and arrangement of hepatocytes transduced with PLV-C/EBPβ appeared more regular and orderly, and intercellular connection was more compact (Figure 2C). In addition, the expression of epithelial marker E-cadherin (E-cad) was upregulated, and bile duct marker CK7 was down-regulated in the HPCs transduced with PLV-C/EBPβ (Appendix A). Furthermore, RNA-seq analysis verified phenotypes of hepatocyte genes and bile duct genes (Figure 2F). In terms of functionality, the protein of albumin synthesis and secretion was significantly reinforced (Figure 2H). Overall, these results provided a viewpoint that the enhanced expression of C/EBPβ could significantly improve hepatic differentiation.

### 3.3. The Overexpression of C/EBPβ Promoted the Proliferation of HPCs

In the aforementioned results, the overexpression of C/EBPβ improved hepatic differentiation; in addition to that, it was shown that C/EBPβ was also elevated to promote cell proliferation. Firstly, the cell morphology showed more dense cell areas on the plate (Figure 2C,G), then, cell count results showed that cell density was higher in HPCs transduced with PLV-C/EBPβ than those with PLV-vector (Figure 3A,B). Next, RNA-Seq results showed that, compared with gene expression in HPCs transduced with PLV-Vector, most of the proliferation-related genes were upregulated in HPCs transduced with PLV- C/EBPβ. To be specific, gene set enrichment analysis (GSEA) revealed that the enrichment plots about cell cycle and DNA chain replication or elongation were enriched in HPCs transduced with PLV-C/EBPβ (Figure 3C). Heatmap of RNA sequencing data indicated that a large number of proliferation genes were significantly upregulated in HPCs transduced with PLV-C/EBPβ (Figure 3D). On the contrary, proliferation suppressor genes, such as E2F8, THAP11, and HECA, showed opposite expression patterns (Figure 3D). KEGG pathway analysis, combined with the gene ontology (GO) functional classification and enrichment analysis of differentially expressed genes (DEGs), further provided evidence that cell cycle and DNA replication pathways were significantly enriched in HPCs transduced with PLV-C/EBPβ (Figure 3E and Appendix A). Additionally, the volcano map of DEGs showed that proliferative genes such as CDC45L, CDC25C, PCNA, E2F1, and E2F2 were significantly upregulated in HPCs transduced with PLV-C/EBPβ (Figure 3F). Furthermore, RT-qPCR analysis of proliferation gene expression showed that several crucial genes such as PCNA, CDC45L, CDC25C, MCM3, GINS1, and CCND1 were significantly increased with the enhanced C/EBPβ expression (Figure 3G). Notably, gene sets associated with E2F targets showed obvious enrichment plots in HPCs transduced with PLV-C/EBPβ (Figure 3C). The members of the E2F family are classical transcription factors related to cycle regulation, and many genes related to cell proliferation and cell cycle are also known as E2F regulatory target genes [29]. Gene quantitative assay also determined that the expression of E2F1 and E2F2 was elevated after the enhanced C/EBPβ expression (Figure 3H). These results demonstrated that the enhanced expression of C/EBPβ could significantly increase proliferative phenotype. To be more precise, the expression of proliferation genes was markably promoted, and cell-cycle-related pathways were significantly enriched due to C/EBPβ overexpression in HPCs.

### 3.4. C/EBPβ Knockdown Impaired Hepatic Differentiation and Blocked HPC Proliferation

It has been proved above that the overexpression of C/EBPβ facilitated hepatic differentiation and accelerated the proliferation of HPCs. In order to further confirm the regulatory characteristics by C/EBPβ, siRNA (C/EBPβ)-mediated gene silencing of C/EBPβ in HPCs on days 7 and 12 were performed, respectively (Figure 4A). Total RNAs were collected 2 days after transductions with siRNAs at two time points, and subsequently, qRT-PCR was performed. On day 9 (2 days post gene silence on day 7), despite C/EBPβ knockdown, there was no significant down-regulation of hepatocyte genes such as HNF4A and AFP, except the expression of ALB showed was slightly decreased (Appendix A). Consistently, there was no significant difference in cell morphology (Appendix A). However, on day 14 (2 days post gene silence on day 12), the cell morphology was changed, cell density was decreased, and the decline of tight junctions in the cells transduced with si-C/EBPβ when compared with those in cells transduced with si-NC (Figure 4B). Obviously, with the efficient knockdown of C/EBPβ on day 14 (Figure 4C), the secretion of ALB was reduced robustly (Figure 4D). qRT-PCR results revealed that hepatic-related genes, including AFP, ALB, ASGPR1, A1AT, HNF4A, and C/EBPα were significantly down-regulated (Figure 4E). In the aforementioned results, the overexpression of C/EBPβ could significantly promote cell proliferation. Those results inspired us to determine if the expression of proliferation-related genes was affected during the C/EBPβ knockdown. As expected, the expression of proliferative genes CDC25C, CDC45L, and MCM3 was significantly down-regulated, and the transcription factor E2F2, which positively regulates proliferation, was also significantly down-regulated (Figure 4H). Moreover, heat map visualization of RNA-seq data showed that the expressions of hepatocyte genes and proliferative genes were significantly down-regulated after the C/EBPβ knockdown (Figure 4F), and GSEA results showed that gene sets related to MYC targets, cell-cycle literature, mitosis targets, and E2F binding were significantly enriched in Si-NC control group (Figure 4I), indicating that the knockdown of C/EBPβ negatively regulated cell-cycle-related pathway genes. Moreover, KEGG pathway enrichment analysis also exhibited that the cell-cycle pathway was enriched by si-NC than those by si-C/EBPβ (Appendix A). Additionally, cell number was significantly decreased when the knockdown of C/EBPβ by siRNA also occurred in the fetal liver cell line LO2 cells (Figure 4G and Appendix A). These results demonstrated that the knockdown of C/EBPβ significantly inhibited the differentiation, proliferation, and cell-cycle progression in HPCs, and also revealed that C/EBPβ was essential for hepatic differentiation and proliferation.

### 3.5. C/EBPβ Coupled with E2F2 Orchestrated Cell Proliferation of HPCs

To further elucidate the mechanism that C/EBPβ regulated the proliferation of HPCs, cell-cycle-related proteins, and pathways affected by C/EBPβ or coupled proteins bound to C/EBPβ were detected by Western blot. First of all, the efficiency of knockdown (si-RNAs) or overexpression (PLV-C/EBPβ) of C/EBPβ was determined on day 14 after the hepatic differentiation (Figure 5A), and the statistical results showed quantitative patterns (Figure 5B). In terms of proliferation, it was found that the expression of several proliferative proteins, including PCNA, CDC45L, and E2F2, were down-regulated after the knockdown of C/EBPβ (Figure 5C,D), and the results were consistent with the qRT-PCR analyses described above (Figure 4H). On the contrary, the above-mentioned proteins, mainly CDC45L and E2F2, were significantly upregulated after overexpression of C/EBPβ (Figure 5E,F). In order to further reveal the mechanism by which C/EBPβ regulated the proliferative phenotype, multiple approaches were conducted. Firstly, differentially expressed gene analysis using RNA-seq data showed that the PI3K/AKT/mTOR signaling pathway was significantly enriched with overexpression of C/EBPβ (Figure 3C). PI3K/AKT/mTOR signaling pathway was reported to regulate the differentiation of hESC-derived hepatocytes in the presence of an insulin signal [30]. In addition, another study found that HGF activated C/EBPβ and cell replication via the PI3K signaling pathway in rat hepatocellular carcinoma cells [31]. Therefore, these results encouraged us to explore whether C/EBPβ could promote cell proliferation by regulating PI3K/AKT/mTOR signaling pathway in HPCs. However, Western blot results showed that knockdown of C/EBPβ hardly changed the expression of AKT and p-AKT (Appendix A) and mTOR and p-mTOR proteins (Appendix A) on day 14, indicating that the regulation of cell proliferation by C/EBPβ was not mediated through PI3K/AKT/mTOR signaling pathway. Consistent with the results of overexpression/knockdown of C/EBPβ, we found that the expression of E2Fs showed the same trends of C/EBPβ (Figure 3G and Figure 4H,I). GSEA results also exhibited that gene sets associated with E2F binding and E2F targets were significantly enriched once C/EBPβ existed (Figure 3A and Figure 4I). E2Fs have been reported as classical cycle-regulated transcription factors, activated by C/EBPβ and its coactivator CREB-binding protein/P300 in vivo [29]. It has been reported that the E2Fs family contains eight known E2F proteins (E2F1-8); E2F1 to E2F3 accumulate in the G1 phase and play an important role in promoting the expression of S-phase target genes. In contrast, E2F4–E2F8 act as transcriptional repressors, in particular for E2F7 and E2F8, which come from a distinct branch of the E2F network involved in repressing S-G2 transcription, providing an important constraint on E2F1 hyperactivation [32]. In our results, we found that the expression of the E2F2 protein was changed most significantly after C/EBPβ expression had been downregulated. Logically, it was worth speculating whether C/EBPβ and E2F2 had direct binding events. Then, Co-IP assays were performed to further investigate the direct binding events between the two proteins, and the result showed that C/EBPβ could directly bind with E2F2 protein in HPCs, and this result was further verified in LO2 cells, a hepatic progenitor cell line (Figure 5G,H). These results indicated that C/EBPβ was required for the expression of proliferative proteins and coupled with cell-cycle-regulated transcription factor E2F2 together by direct binding to promote the proliferative phenotype.

### 3.6. C/EBPβ Promoted the Expression of Proliferative Genes by Binding to Promoter Regions

As a transcription factor, whether C/EBPβ was capable of binding to the promoter or enhancer regions of proliferative genes to promote gene expression, Cut&Tag projects were performed to verify our speculation. Cut&Tag (Cleavage Under Targets and Tagmentation), as an emerging technology, was able to replace the traditional ChIP-seq to explore the genomic DNA sequences bound to the target protein. Compared with ChIP-seq, Cut&Tag showed many advantages, including low background noise, less cell demand, and high data repeatability [33]. As a result, Cut&Tag was conducted on day 14 to find the DNA sequences bound by C/EBPβ. Firstly, the information on peaks was analyzed, and we found that the length of all peaks was mostly confined to 300–600 bp (Appendix A), and the distribution of peaks was concentrated near the transcription start site (TSS) of genes (Figure 6A and Appendix A). Next, the functional components of peak distribution revealed that most peaks were located in promoters, distal regulatory regions, and introns (Figure 6B). The above peak distribution was basically consistent with the binding characteristics of transcription factors. Moreover, there were many common motif sequences derived from the enriched DNA peaks, and the top motif sequences were highly similar to the motif sequences bound by C/EBP family proteins (Appendix A), indicating that the peak information obtained by sequencing was indeed the binding site of C/EBPβ. Furthermore, the most representative motif sequences were ‘ATTGCACAACA’ (Figure 6C). In this study, we aimed to find enriched binding regions of transcription factor C/EBPβ on the whole genome landscape. Intriguingly, peaks from C/EBPβ bindings showed significant enrichment pathways associated with the cell cycle (Figure 4D and Appendix A), and individual analysis of a single gene demonstrated that proliferative genes were also exhibited in the top lists. Representative chromatin landscapes across 2–20 Mb segments of the human genome generated by IGV software and the regulatory regions of multiple proliferative genes showed the distinct location and peak heights. Peaks from individual proliferative gene, including CDC25C, CDC45L, PCNA, CDC7, and E2F2, were enriched in the transcriptional regulatory regions of their respective genes, and the height of the peaks of these genes was increased after the enhanced expression of C/EBPβ (Figure 6E–I). These results indicated that C/EBPβ bound to the promoter region of proliferative genes to boost cycle progression in HPCs cells.

### 3.7. Cell-Cycle Progression was Promoted by C/EBPβ in LO2 Cells

Next, in order to verify the results of C/EBPβ binding to promoter regions of proliferative genes from CUT&Tag data, a luciferase activity assay was performed. Firstly, the prediction of binding sites showed three major binding sites of C/EBPβ and selected pivotal proliferative gene CDC45L promoter (Figure 7A). Then, the luciferase activity assay was carried out, and we found that the luciferase activity was significantly enhanced after co-transduction of C/EBPβ plasmid with CDC45L plasmid into 293T cells and LO2 cells (Figure 7B), indicating that C/EBPβ bound the promoter of CDC45L gene and enhanced the expression of the luciferase. The above studies have fully demonstrated that C/EBPβ regulated the proliferative phenotype in HPCs by binding the promoter regions of proliferative genes; thus, we speculated whether the same regulatory mechanism existed in hepatic progenitor cell line, LO2 cells. Therefore, Sh-RNA (C/EBPβ) lentiviruses were used (Appendix A) to transduce LO2 cells, and the results showed that the expression of several proliferative genes and E2F transcription factor genes, including CDC25C, CDC45L, E2F1, and E2F2 were decreased after lentivirus-mediated C/EBPβ knockdown (Figure 7C,D). Additionally, cell-cycle progression detected by flow cytometry and corresponding statistical data showed that the proportion of the G1 phase was increased and the S phase was decreased significantly after the knockdown of C/EBPβ in LO2 cells, indicating that cell-cycle progression was inhibited (Figure 7E,F). These results indicated that C/EBPβ also regulated the expression of proliferation genes and cell-cycle phenotype in LO2 cells. Altogether, these results indicated a working mechanism that C/EBPβ initiated to bind the promoter of the proliferation-related transcription factor E2F2 and activate E2F2 expression and then coupled with E2F2 to promote the expression of cell-cycling genes and proliferative genes in hepatic progenitor cells by binding the promoter regions of these genes rather than through regulating PI3K/AKT/mTOR pathway (Figure 7G).

## 4. Discussion

In order to obtain functional hepatocytes for clinical application and pharmaceutical study, a variety of strategies have been reported to generate hepatocytes. Deng et al. [5] reported that human primary hepatocytes could be maintained in vitro with five chemicals (5C) and retained initial hepatic morphology and functional characteristics under prolonged culture. However, primary hepatocytes cultured in this approach were still difficult to expand in vitro, and the numbers were still limited. To expand the number of hepatocytes, some studies have focused on reprogramming primary hepatocytes into hepatic progenitor cells after massive expansion at this stage, which was re-differentiated into mature functional hepatocytes [34,35,36]. Although HPV E6/E7 or SV40 large T antigen can improve the proliferation potential of hepatic progenitor cells, the immortalization mediated by the overexpression of these oncogenes also has the risk of tumorigenesis, limiting their applications. In contrast, pluripotent stem cell-derived hepatic progenitor cells or hepatoblasts may have an advantage because these cells are stable in origin, have defined differentiation routes, and can effectively differentiate into mature hepatocytes. However, the number of hepatocytes derived from pluripotent stem cells is still limited by the initial cells, and the resulting hepatoblasts or hepatocytes exhibit limited proliferation potential. In order to find efficient and economical methods to generate a large number of high-quality functional hepatocytes, Li’s group established a cost-effective culture condition that enabled long-term in vitro expansion of purified hepatoblasts and maintained the dual differentiated potency [9]. The expanded hepatoblasts could be differentiated into hepatocytes with robust hepatic function characteristics in vitro.

Although hepatoblast proliferation has been well studied, the endogenous regulatory mechanisms involved in hepatocyte production and proliferation are still poorly understood. In our study, the initial results demonstrated that C/EBPβ played an important role in regulating hepatocyte differentiation from hESCs (Figure 2E,H). Subsequently, RNA-Seq data indicated that genes and signaling pathways associated with cell cycle and cell proliferation were significantly enriched with overexpression of C/EBPβ (Figure 3A,C,F). Moreover, our further results revealed that the transcription factor C/EBPβ could directly bind to promoter regions of proliferative genes such as CDC45L, CDC25C, CDC7, and PCNA (Figure 6E–I), and the binding activities were validated by the luciferase assay (Figure 7B).

The transcription factor C/EBPβ is a member of the CCAAT enhancer binding protein family, which has six members, including C/EBPβ [37]. It was reported that C/EBPβ plays important roles in differentiation, energy metabolism, cell cycle, migration, and invasion of a variety of cells [11,12,13]. For instance, C/EBPβ was rhythmically expressed in the liver and regulated by physiological rhythm and nutritional signals, thereby regulating the circadian autophagy process in the liver [15]. In hepatectomy mice, the expression of C/EBPβ was rapidly increased and involved in the entire process during the liver regeneration, and dynamic profiles of binding analysis by ChIP-seq further revealed that C/EBPβ played specific regulation of cell-cycle genes during the liver regeneration [16]. In adipocytes, C/EBPβ was induced at the early stage of differentiation and activated the expression of CEBPα and PPARγ by directly binding to the promoter region, which promoted adipocyte proliferation and differentiation [38]. In preadipocytes, the expression of four cycle genes CDC45L, MCM3, GINS1, and CDC25C was induced by C/EBPβ, which promoted preadipocytes to re-enter the cell cycle [19]. In mammary cell differentiation, it was essential that C/EBPβ regulated the development, differentiation, and proliferation of mammary epithelial cells [39].

The specific mechanism of proliferation regulated by C/EBPβ was reported in an earlier study. Greenbaum et al. [29] reported that C/EBPβ occupied promoters of E2F-regulated growth-related genes and promoted cell-cycle progression in an in vivo mammalian model, which identified C/EBPβ as a direct activator of E2F target genes during cell-cycle progression, and the mechanism involved CBP/P300 recruitment [29]. E2Fs are classical transcription factors that play an important role in the regulation of cell-cycle transition, in which E2F1-E2F3 play an important role in promoting the expression of target genes in the S phase [32]. In our study, E2F2 was the most principal target regulated by C/EBPβ (Figure 3G and Figure 4I) and was also the only C/EBPβ binding site in the promoter region of E2F family transcription factors (Figure 6I and Appendix A). Additionally, the Co-IP protein pull-down assay also further demonstrated that C/EBPβ could bind to E2F2 (Figure 5G,H). Collectively, these results demonstrated that C/EBPβ first activated E2F2 and then coupled with E2F2 bound to the promoter regions of cell proliferative genes to promote cell proliferation in hESC-derived hepatic progenitor cells (Figure 7G). Thus, our findings open a new avenue to provide an in vitro efficient approach to generate proliferative hepatocytes to potentially meet demands for use in cell-based therapeutics as well as a test system for pharmaceutical and toxicological studies.

## 5. Conclusions

In conclusion, our results demonstrated that C/EBPβ first activated E2F2 and then coupled with E2F2 bound to the promoter regions of cell proliferative genes (CDC45L, CDC25C, and PCNA) to promote those gene expressions and to accelerate cell proliferation of hESC-derived hepatic progenitor cells. Additionally, the findings of the mechanism were verified in LO2 cells, a hepatic progenitor cell line.

## Figures and Tables

**Figure 1 cells-12-00497-f001:**
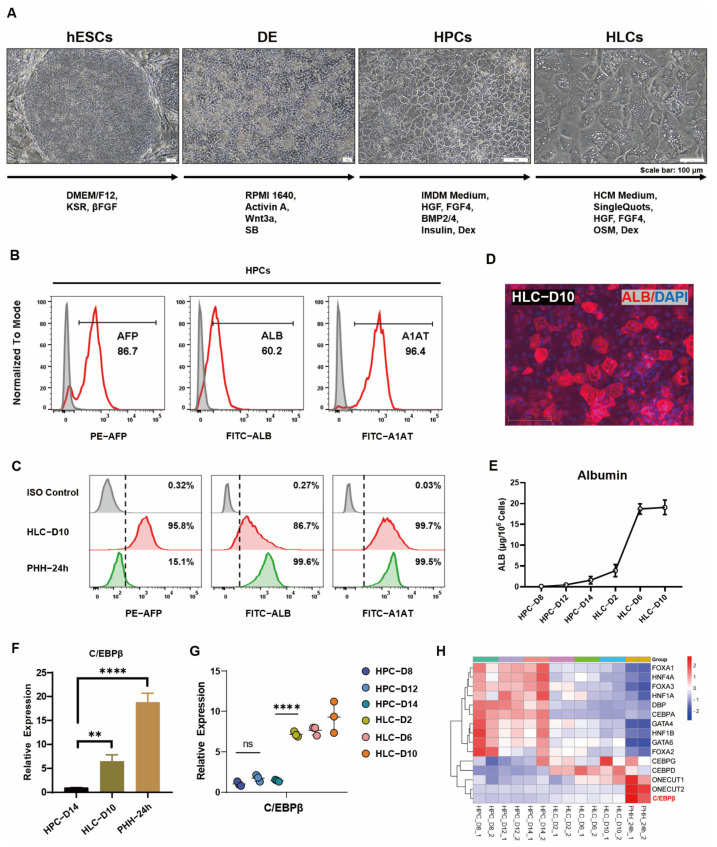
C/EBPβ was enriched in liver and correlated with hepatocyte differentiation. (**A**) Schematic diagram of the stepwise induction of hESCs into hepatocyte-like cells (HLCs), with representative images showing the morphology of the cells at sequential stages of hESCs, DE cells, HPCs, and HLCs, respectively. Scale bar, 100 μm. (**B**) Flow cytometry analysis for the expression of specific markers at stage of HPC-D14 (AFP, ALB, A1AT). (**C**) Flow cytometry analysis for the expression of hepatocellular markers at HLC-D10 and PHH-24 h (AFP, ALB, A1AT). (**D**) The expression of ALB was determined by immunofluorescence staining of HLC on day 10. (**E**) Albumin secretion was quantified by taking advantage of ALB ELISA Kit at sequential stages of HPCs and HLCs. (**F**) The expression of C/EBPβ was upregulated along the cell maturation process. (**G**) The dynamic expression of C/EBPβ at different time points during the hepatic differentiation was determined by qPCR. (**H**) Heat map visualization of RNA-Seq data indicated that selected liver-enriched transcription factors (LETFs) were expressed in our hepatic differentiated cells and in primary human hepatocytes. Data represent mean ± SEM with *p*-values indicated when significant. *p* < 0.01 **, *p* < 0.0001 ****, ns, not significant. Data are the three samples per group.

**Figure 2 cells-12-00497-f002:**
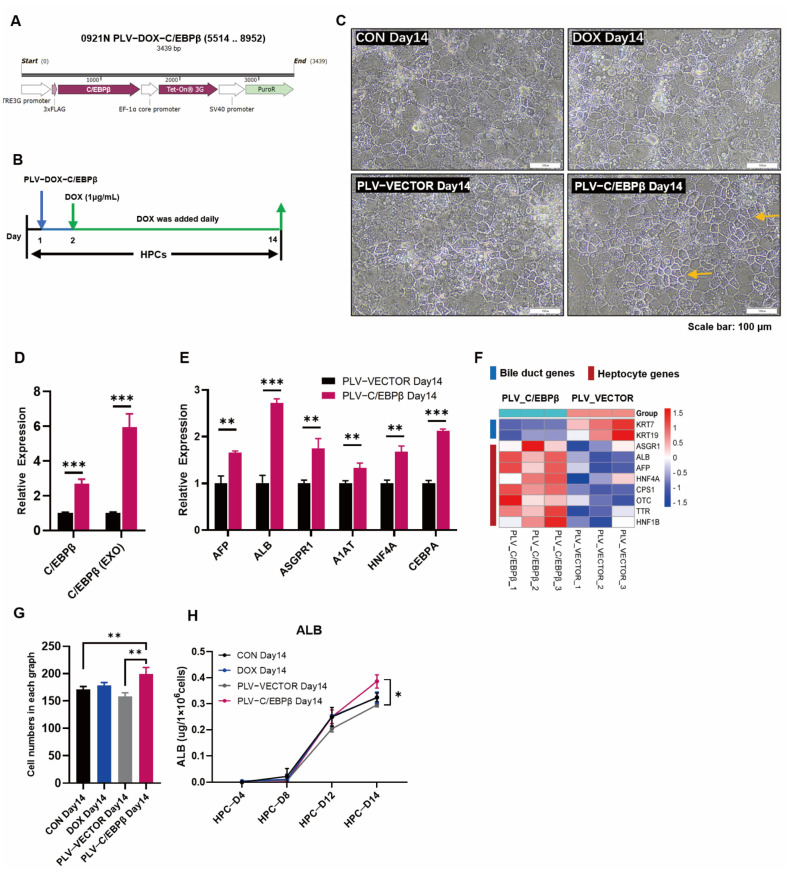
Enhanced expression of C/EBPβ promoted hepatic differentiation. (**A**) Schematic illustration of lentivirus vectors with overexpression of C/EBPβ. (**B**) Lentivirus vector-mediated overexpression of C/EBPβ by the administration of DOX during the hepatic differentiation. (**C**) Representative images of cell morphologies under different treatments, dense areas of cells are indicated by yellow arrows. (**D**) The overall C/EBPβ expression and exogenous C/EBPβ expression after the transduction with PLV-C/EBPβ virus and DOX administration (1 μg/mL). (**E**) Relative expression analysis of hepatocyte genes including AFP, ALB, ASGPR1, A1AT, HNF4A, and CEBPα in HPCs transduced with PLV-Vector or PLV-C/EBPβ on day 14. (**F**) Heat map visualization of RNA-Seq data indicated that selected bile duct genes and hepatocyte genes were differentially expressed between HPCs transduced with PLV-Vector or PLV-C/EBPβ on day 14. (**G**) Cell numbers were counted in each group in (**C**) and compared between different treatments. (**H**) ALB secretion was determined from the supernatants of HPCs with different treatments at different time points during the differentiation. Data represent mean ± SEM with *p*-values indicated when significant. *p* < 0.05 *; *p* < 0.01 **, *p* < 0.001 ***.

**Figure 3 cells-12-00497-f003:**
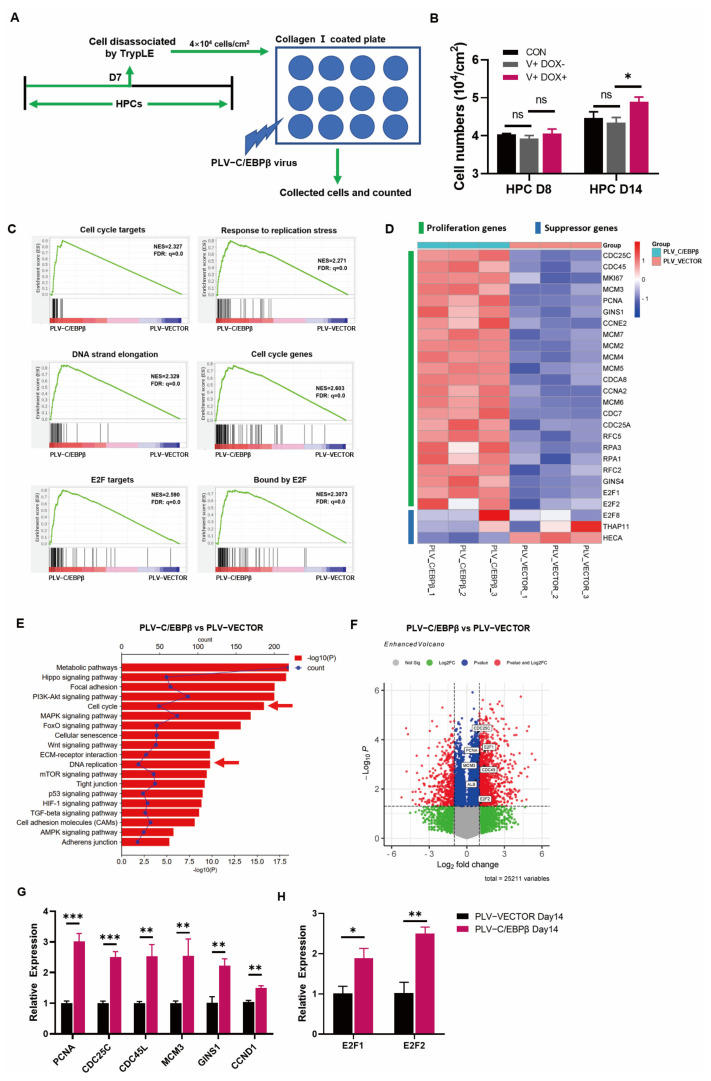
The overexpression of C/EBPβ promoted the proliferation of HPCs. (**A**) The diagram shows that cells were disassociated by TrypLE at HPC D7 and then seeded at 4 × 10^4^ cells/cm^2^ on collagen type I coated plates. Simultaneously, PLV-C/EBPβ virus was administrated. Subsequently, cells were disassociated again and counted at HPC D8 and HPC D14, respectively. (**B**) Cell numbers were determined by cell counts for HPC on day 8 and for HPC on day 14. (**C**) GSEA results showed that cell cycle and E2F targets related genes were significantly enriched in HPCs transduced with PLV-C/EBPβ on day 14 when compared with those transduced with PLV-Vector on day 14. (**D**) Heat maps of selected proliferative genes and suppressor genes were differentially expressed between HPCs transduced with PLV-Vector or with PLV- C/EBPβ on day 14. (**E**) The KEGG pathways for the differentially expressed genes between HPCs transduced with PLV-Vector or with PLV- C/EBPβ on day 14. The principal pathways were the cell cycle and DNA replication. (**F**) Volcano plots showed significant changes in cell-cycle-related genes between HPCs transduced with PLV-Vector or with PLV- C/EBPβ on day 14. Volcano plot showed −log10 (*p*-value) on the y-axis versus log2 (fold change) on the x-axis. Each point represented a different gene. (**G**,**H**) Relative expression of proliferative genes, including PCNA, CDC25C, CDC45L, MCM3, GINS1, and CCND1 (**F**) and E2F genes, including E2F1 and E2F2 (**H**) in HPCs transduced with PLV-Vector or with PLV- C/EBPβ on day 14. Data represent mean ± SEM with *p*-values indicated when significant. *p* < 0.05 *, *p* < 0.01 **, *p* < 0.001 ***, ns, not significant. Data are the three samples per group.

**Figure 4 cells-12-00497-f004:**
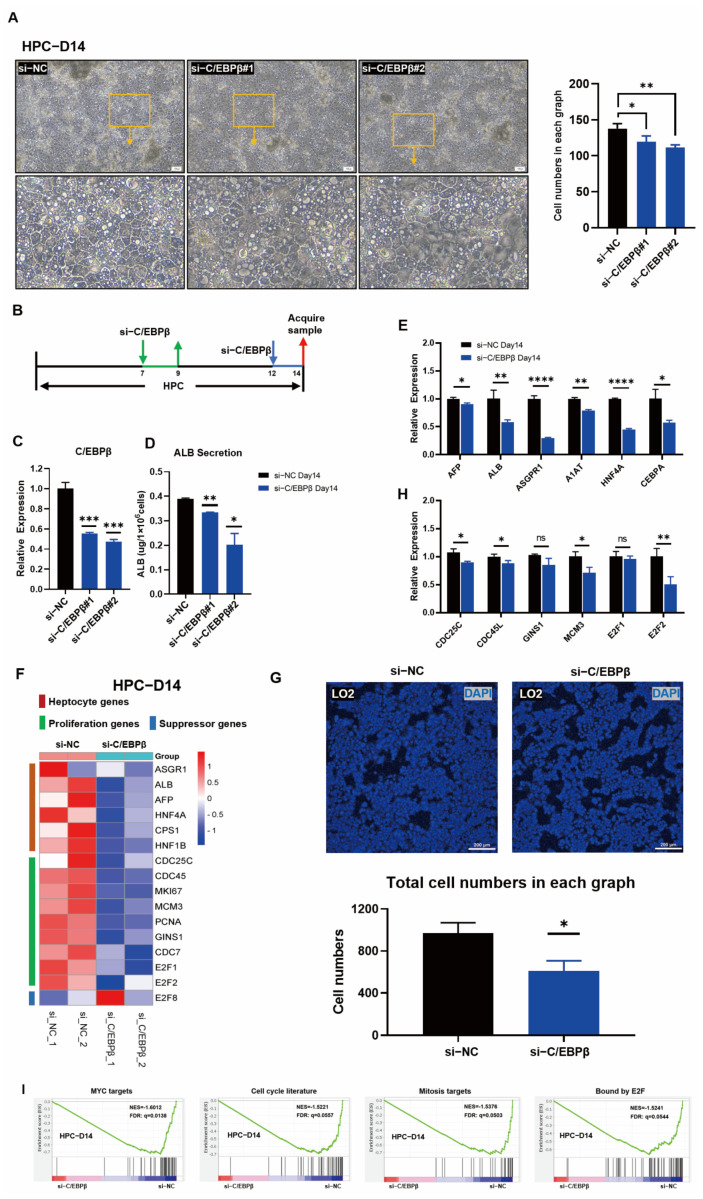
C/EBPβ Knockdown impaired hepatic differentiation and blocked proliferation of HPCs. (**A**) Representative images of cell morphologies of HPCs 2 days after si-RNA treatments on day 12 and cell numbers were counted in each group in (**A**). (**B**) Schematic illustration of si-RNA mediated C/EBPβ knockdown during the hepatic differentiation of hESCs. (**C**) Knockdown efficiency of C/EBPβ was detected by qRT-PCR on day 14 of HPCs. (**D**) ALB secretion was detected from the supernatants of cultured HPCs 24 h after siRNA (C/EBPβ) treatment. (**E**) Relative gene expression analysis of hepatocyte genes including AFP, ALB, ASGPR1, A1AT, HNF4A, and CEBPα in HPCs treated with si-NC or with si-C/EBPβ on day 14. (**F**) Heat map of selected genes, including hepatocyte genes, proliferative genes, and suppressor genes in HPCs treated with si-NC or with si-C/EBPβ on day 14. (**G**) The same number of LO2 cells were seeded, and then si-RNA mediated C/EBPβ knockdown was conducted. After 3 days of treatments, DAPI staining was performed to mark the nuclei, and then cell numbers were defined by ImageJ software. (**H**) Relative expression of proliferative genes including CDC25C, CDC45L, GINS1, MCM3, E2F1, and E2F2 in HPCs treated with si-NC or with si-C/EBPβ on day 14. (**I**) GSEA results showed that cell-cycle-related gene sets, such as MYC targets, cell-cycle literature, and mitosis targets bound by E2F, were significantly enriched in HPCs treated with si-NC when compared with those in HPCs treated with si-C/EBPβ. Data represent mean ± SEM with *p*-values indicated when significant. *p* < 0.05 *, *p* < 0.01 **, *p* < 0.0001 ****, ns, not significant. Data are the three samples per group.

**Figure 5 cells-12-00497-f005:**
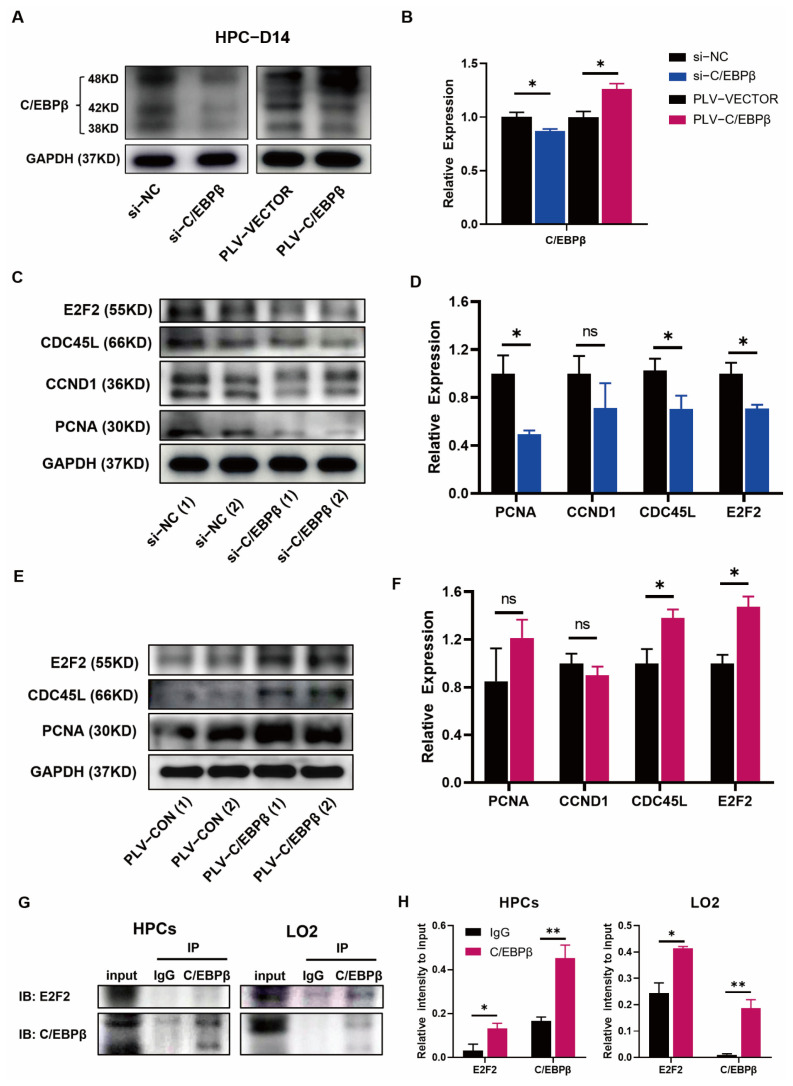
C/EBPβ coupled with E2F2 orchestrated cell proliferation of HPCs. (**A**,**B**) Western blot analysis of C/EBPβ on day 14 of HPCs after the knockdown or overexpression of C/EBPβ (**A**), and the statistical results (**B**). (**C**,**D**) Western blot analysis of representative protein expressions associated with cell proliferation after knockdown of C/EBPβ **(C)**, and corresponding statistical results (**D**). (**E**,**F**) Western blot analysis of representative protein expressions associated with cell proliferation after overexpression of C/EBPβ (**E**) and corresponding statistical results (**F**). (**G**,**H**) Co-immunoprecipitation (Co-IP) assays to validate interaction of C/EBPβ with E2F2 in HPCs on day 14 and LO2 cells. The cell lysates were subjected to immunoprecipitation with C/EBPβ antibody, and the resulting immunoprecipitants were analyzed in the immunoblot with E2F2 and C/EBPβ antibodies (IB: E2F2; IB: C/EBPβ). Input was also subjected to immunoblot to represent a positive control, and non-specific IgG represented as a negative control. Data represent mean ± SEM with *p*-values indicated when significant. *p* < 0.05 *, *p* < 0.01 **, ns, not significant. Data are the three samples per group.

**Figure 6 cells-12-00497-f006:**
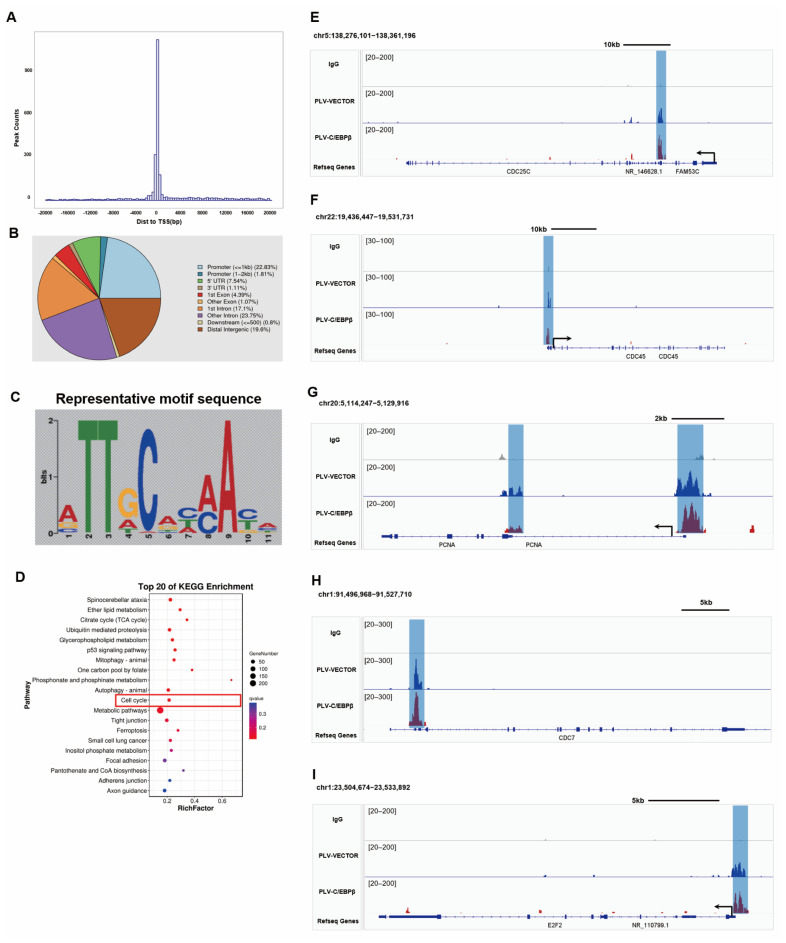
C/EBPβ promoted the expression of proliferative genes by binding to promoter regions. (**A**) CUT&Tag data analysis for peak summit of relative genes at location map, the abscissa was the distance between peak summit and TSS site, and the ordinate was the number of peaks. (**B**) Pie chart of peak distribution on gene functional elements. (**C**) Top scoring and representative motif sequence of C/EBPβ binding sites. (**D**) KEGG enrichment bubble diagram to show the first 20 pathways with the smallest q values, pathway shown as the ordinate and rich factor shown as the abscissa (the number of differences in this pathway divided by all the numbers). The size indicated the number; the redder the color, the smaller the q value. (**E**–**I**) The peaks of individual proliferative genes, including CDC25C, CDC45L, PCNA, CDC7, and E2F2, were enriched in the transcriptional regulatory regions of their respective genes, and the peaks of some genes were enhanced after overexpression of C/EBPβ.

**Figure 7 cells-12-00497-f007:**
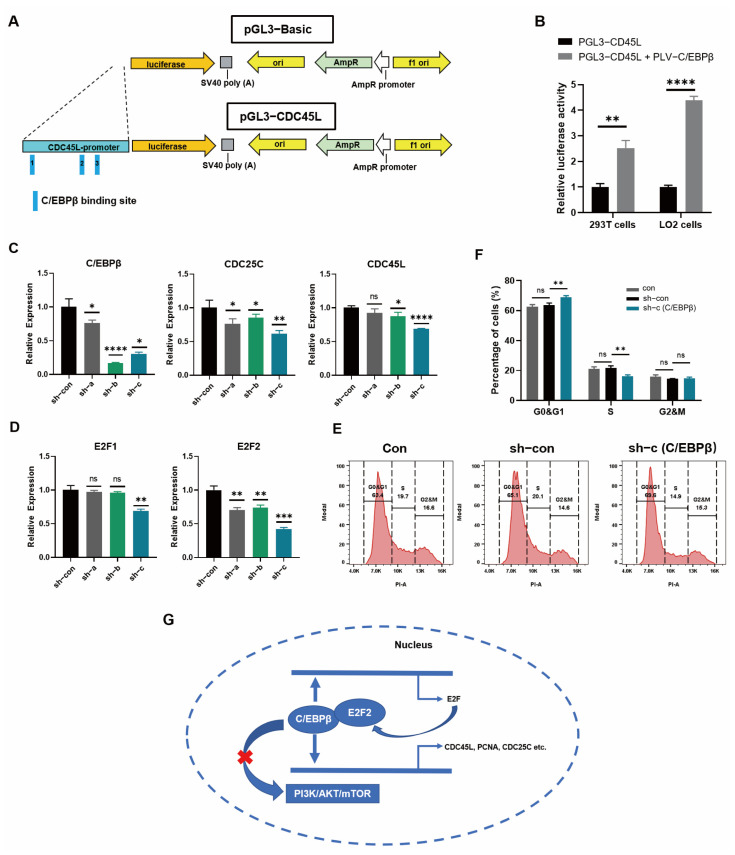
Cell-cycle progression was promoted by C/EBPβ in LO2 cells. (**A**) The schematic diagram shows that C/EBPβ binds to CDC45L promoter regions to regulate luciferase expression. (**B**) Luciferase assays were carried out to verify the C/EBPβ binding activity at promoter regions of CDC45L gene in 293T cells and LO2 cells. (**C**,**D**) Lentivirus-mediated C/EBPβ knockdown with three pairs of sh-RNA (sh-a, sh-b, and sh-c) was performed to determine the expression of cell-cycle genes (**C**) and transcriptional factor E2Fs genes (**D**) by qRT-PCR assays. (**E**,**F**) Flow cytometry was performed to detect the proportion of cell-cycle phases after the treatment with sh-RNA (C/EBPβ) in LO2 cells (**E**), and the statistical results of proportion were depicted (**F**) by GraphPad Prism 8 software. (**G**) C/EBPβ coupled with E2F2 regulated cell-cycle gene expression in hepatic progenitor cells. C/EBPβ first bound to the regulatory region of cell-cycle-regulating transcription factor E2F2 to activate E2F2, then C/EBPβ coupled with E2F2 to promote the expression of cell-cycle genes (CDC45L, CDC25C, and PCNA) in hepatic progenitor cells. Data represent mean ± SEM with *p*-values indicated when significant. *p* < 0.05 *, *p* < 0.01 **, *p* < 0.001 ***, *p* < 0.0001 ****, ns, not significant. Data are the three samples per group.

## Data Availability

The data that support the findings of this study are available from the corresponding author upon reasonable request.

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
