# Peer review of "C/EBPβ Coupled with E2F2 Promoted the Proliferation of hESC-Derived Hepatocytes through Direct Binding to the Promoter Regions of Cell-Cycle-Related Genes"

_cells, 2023, doi:10.3390/cells12030497_

Round 1

Reviewer 1 Report

This paper described the discovery of C/EBPβ coupled with E2F2 through the direct binding to the promoter regions of cell cycle-related genes to promote the proliferation of human embryonic stem cell (hESC)-derived hepatocytes. The paper is technically sound, employs different techniques, and could be of interest to the Cells, MDPI community. However, in some parts, the paper is confusing and therefore needs revision before being considered for publication.

1

11.      Please remove the red line underneath of hESCs in Figures 1A, 1B, 1D, and Figure 4F (hepatocytes & suppressor gene), as well as Figure 7A (ori & AmpR).

2.       Please standardize to use C/EBPβ, in all the graphs (especially Figure 1E&IF) and throughout the manuscript instead of CEBPB or CEBPβ.

3.       Please specify the “ ****” in the Figure legends of 1E, 1F 4E 7B, 7C, respectively, as well as in the “ 12 Statistics” under the session of “2. Materials and Methods”.

4.       Authors describe “…Briefly, hESCs mainly 253 underwent three stages to differentiate into hepatocyte-like cells (HLCs), and the resulting 254 HLCs showed typical polygonal characteristics in morphology (Figure 1A)...” (page 6, line 253-255).

Authors should perform & include immunostaining to show the spatial organization of the hepatic markers after differentiation of hESC, or some functional tests (albumin, CYPs, urea, etc) to prove this statement, rather than just showing the brightfield images which are not very convincing. Also, please enlarge the brightfield images of Figure 1A, so that the detail of the cell morphology can be more visible.

5.       Figure 1 legend, line 272: A full stop needs to be included before the “Data”.

6.       Similarly, authors should provide immunostaining images with quantification data for Figure 2C to prove/claim that the overexpression of C/EBPβ does change the morphology and compactness of the hepatocytes. The current brightfield image in Figure 2C is not enough to prove it unless the cell morphology or cell compactness can be quantified from these images.

7.       Figure 2H: The quantifications presented in Figure 2H lack error bars and statistical significance testing. Were these albumin levels performed on n=1 HPCs samples? Without more convincing data on quantitative differences in albumin secretion, many claims with respect to hepatic differentiation within the manuscript need to be mitigated or better justified because albumin is one of the key functional tests for hepatocytes.

8.       Figure 3E: Authors did quantify the RNA contents to show the higher cell density. However, the RNA content isn’t a standard test to show the cell density of HPCs. Flow cytometry or other methods should be used to support the RNA content to prove the increment of cell density.

9.       Authors only focus on studying the gene expression/transcriptional level of the hepatic differentiation and the proliferation of HPCs. Besides the transcriptional expression, it is important to include protein expression data to support the claim on the hepatic differentiation and HPCs proliferation.

10.   It is also important to quantify the number of functional hepatocytes (albumin test, urea test, CYP test) to support/claim that the differentiated hepatocytes are functional.

11.   Figure 4C & 4D: Authors should provide western blot data for C/EBPβ knockdown & control. A gene expression of C/EBPβ is not representative enough to prove that C/EBPβ has been knock-downed from the cell.

12.   Figure 4B: Please provide the immunostaining data with quantification graphs to support your claim that C/EBPβ knockdown impairs the hepatic differentiation and block the HPC proliferation.

13.   Figure 5H: Please add the statistical analysis and error bar in Figure 4H.

Reviewer 2 Report

In the presented manuscript, Liu and her colleagues confirmed C/EBPβ coupled with E2F2 promoted the proliferation of hESC derived hepatocytes through directly binding to the promoter regions of cell cycle-related genes. The researcher have provided solid data about that overexpression of C/EBPβ can drive hESC to elevate hepatic phenotype in vitro, and the results are interesting, but it does not mean the effect is large.

1.There is still a big gap between the obtained cells and the primary hepatocytes in terms of gene expression and function from Figure 1E, 1F. It is unknown whether the uncomplete differentiation due to the single transcription factor C/EBPβ or not the best differentiation time, because C/EBPβ is also elevated to promote cell proliferation. 

2. I'm a little confused that overexpression of C/EBPβ promoted cell differentiation in Figure 2, but overexpression of C/EBPβ promotes HPC proliferation in Figure 3. After overexpression of C/EBPβ, is the cell in the HPC stage or in the HLC stage? 

3. As we all known, primary hepatocytes with well differentiated and lower proliferation capacity so they are the best cell source for cell therapy. Therefore, I think the cells provided by the author are still far from the cells of therapeutic grade. 

However, the author reveals that C/EBPβ first initiated E2F2 expression, then coupled with E2F2 to regulate the expression of proliferative genes in hepatocytes during the differentiation of hESCs, revealing a novel regulatory mechanism of C/EBPβ during the differentiation of ESCs to hepatocytes. 

Round 2

Reviewer 1 Report

It is good to be published. 
